# Local Constraint-Based Causal Discovery under Selection Bias

**Philip Versteeg**        P.J.J.P.VERSTEEG@UVA.NL
*Informatics Institute*
*University of Amsterdam*

**Cheng Zhang**        CHENG.ZHANG@MICROSOFT.COM
*Microsoft Research Cambridge*

**Joris M. Mooij**        J.M.MOOIJ@UVA.NL
*Korteweg-de Vries Institute*
*University of Amsterdam*

**Editors:** Bernhard Schölkopf, Caroline Uhler and Kun Zhang

## Abstract

We consider the problem of discovering causal relations from independence constraints selection bias in addition to confounding is present. While the seminal FCI algorithm is sound and complete in this setup, no criterion for the causal interpretation of its output under selection bias is presently known. We focus instead on local patterns of independence relations, where we find no sound method for only three variable that can include background knowledge. Y-Structure patterns (Mani et al., 2006; Mooij and Cremers, 2015) are shown to be sound in predicting causal relations from data under selection bias, where cycles may be present. We introduce a finite-sample scoring rule for Y-Structures that is shown to successfully predict causal relations in simulation experiments that include selection mechanisms. On real-world microarray data, we show that a Y-Structure variant performs well across different datasets, potentially circumventing spurious correlations due to selection bias.

**Keywords:** causal discovery, causal inference, observational and experimental data, selection bias

## 1. Introduction

The discovery of causal relations from data is central to many disciplines in science such as biology, economics and psychology. Information about the true underlying causal relations is fundamental in predicting the effects of new unseen interventions, and direct experimentation is often expensive, unethical or unfeasible. Algorithms for discovering causal relations have been developed for data with a variety of challenging attributes, for example allowing for latent confounding or cyclic causal relations to be present (Spirtes et al., 1999; Mooij and Claassen, 2020).

One of the more demanding properties of data is the presence of selection bias: the selective exclusion of samples in the data-generating process. The biased data suffers from spurious correlations that can severely hinder methods for statistical and causal inference. In practice the possibility of a selection mechanism is often dismissed as a hypothesis. Consider the following example.

**Example 1** *(Gene Regulatory Network) Genes in a* gene regulatory network *(GRN) influence other genes through a process of gene expression. In a microarray experiment, cells of an organism are grown for multiple generations to produce sufficient genetic material, after which expression levels are measured for all genes. Due to exponential growth, the fittest cells quickly dominate*

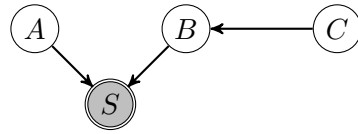

Figure 1: **(Example)** Causal graph accompanying Example. 1, where selection bias is represented by the unobserved and conditioned variable $S$.

*the population, and the measured expressions reflect those of the fittest subpopulation while mostly ignoring those with a slower growth rate.*

*A toy model for only three genes $A, B$ and $C$ is depicted in Fig. 1, where the unobserved selection bias variable $S$ represents the survivability or fitness of the cells. Here, fitness $S$ is a direct effect of the expressions of genes $A$ and $B$, representing multiple causal relations that exist concurrently as a fall-back mechanism in a redundant process.*

*The experiment mostly captures the expressions for genes of the fittest cells, effectively conditioning on the unmeasured $S$ and introducing spurious correlations between $A$ and both $B$ and $C$ in the data. This remains true in post-interventional samples, originating from the knock-out of gene $A$. If selection bias is disregarded, one could conclude from this data that $A$ is an (indirect) cause of both $B$ and $C$. Crucially, these learned spurious causal relations would not represent the true molecular interactions and may transfer poorly to other types of experimental data gathered on the GRN.*

As seen in Example 1, when not accounted for, selection bias can lead to incorrect conclusions when inferring causal relations, which will then transfer poorly to other domains. Additionally, while confounding bias can be excluded through randomization, this is not the case for selection bias, indicating its intrinsic difficulty.

The term 'selection bias' is sometimes used as an umbrella term, indicating a systematic error in the sample that hinders the causal effect estimation and requires adjustment (Bareinboim and Pearl, 2012). In this work we specifically refer to types of selection bias such as non-compliance and volunteer bias (Hernán et al., 2004), that affect the inference of causal relations by changing the independence structure of the data at hand. Presently, there do exist algorithms for causal discovery under selection bias.

The seminal FCI algorithm (Spirtes et al., 1995, 1999) is sound and complete under both confounding and selection bias. It returns a *Partial Ancestral Graph* (PAG), an equivalence class that encodes independence constraints and causal relations, and represents a set of ancestral graphs (Zhang, 2008a). When selection bias is excluded, a criterion for extracting confounding relations from a PAG was introduced by Zhang (2008a). A procedure to read off ancestral relations was first conjectured by Zhang (2006), shown to be sound in Roumpelaki et al. (2016) and generalized to cycles in Mooij and Claassen (2020). However, a systematic procedure for extracting causal relations from this equivalence class when selection bias is present is, to the best of our knowledge, unknown.

**Contributions** In this work, we investigate how several constraint-based causal discovery methods are affected by the presence of selection bias. Empirically, we are motivated by the hypothesis that microarray data is possibly suffering from selection on an unmeasured fitness trait, as described in Example 1.

We take a bottom-up approach and consider methods for causal discovery reason that reason from patterns of (in)dependence relations between three or four variables. These *local* approaches, such as LCD (Cooper, 1997) and (Extended) Y-Structures (Mani et al., 2006; Mooij and Cremers, 2015), have simpler causal semantics that allow for a more straightforward causal interpretation than *global* algorithms such as FCI. Additionally, less statistical testing is required, reducing the risk of error propagation, and these methods have a tractability advantage, trading off the completeness for computational efficiency.

Our contributions are as follows.

- For three variables, we investigate the failure modes of LCD in the presence of selection bias, and find that there exists no three variable constraint-based causal discovery approach if selection bias is admitted. (Section 3.2)

- For four variables, we show that the (Extended) Y-Structures pattern is valid under selection bias and that it predicts an unconfounded causal relation. (Sec. 3.3)

- Using simulations and real-world microarray data, we show a successful application of Y-Structures, implying that a selection mechanism could plausibly underlay the microarray data at-hand. (Sec. 4)

In the methods above we attempt to be as general as possible, and we allow for cycles, latent confounders and multiple selection bias variables to be present.

**Related work**  The recovery from selection bias, where the aim is to infer conditional distributions and causal relations from data under (partial) selection bias when the underlying causal graph is known, has been studied extensively in literature (Bareinboim and Pearl, 2012; Bareinboim et al., 2014; Bareinboim and Tian, 2015; Correa and Bareinboim, 2017; Correa et al., 2018, 2019a,b). Selection bias can be seen as a special case of missingness (Mohan et al., 2013; Tu et al., 2019).

Constraint-based causal discovery methods reason over conditional independence relations derived from data to infer causal relations. The formative PC algorithm (Spirtes et al., 2000) identifies all causal relations from independence statements when all relevant variables have been observed without sample selection. When selection bias or latent confounding are present, the output of the FCI algorithm (Spirtes et al., 1995, 1999) is sound and, augmented with additional rules, complete (Zhang, 2008b). Extensions of FCI include optimizing for computational efficiency (Colombo et al., 2012; Claassen et al., 2013), soundness under interruptions at any stage (Spirtes, 2001) and when cycles are present (Mooij and Claassen, 2020). A sound and complete reformulation of FCI in terms of minimal (in)dependence statements and one additional rule is presented in Claassen and Heskes (2011). In Mooij et al. (2020), a generic framework was introduced for incorporating background knowledge in causal discovery. Cooper (1995) introduced a constraint-based method for detecting selection bias.

In Kemmeren et al. (2014), a microarray dataset with gene expressions is introduced, which has been used for the real-world evaluation of causal discovery methods such as LCD (Versteeg and Mooij, 2019). Meinshausen et al. (2016) shows the successful application of the ICP algorithm Peters et al. (2016) on this dataset.

## 2. Background

We first introduce graphical models, where we allow for cycles and latent confounders, and where we add explicit selection bias variables and context nodes that represent background information.

We write single variables as capitals, i.e. $X$, and sets of variables as boldface, i.e. $\boldsymbol{X}$. In the context of graphical models, we interchangeably use the terms 'variable' and 'node' for a random variable associated to a node.

## 2.1. Causal Graphical Models

A *Directed Mixed Graph* (DMG) is a graph $\mathcal{G} = (\boldsymbol{V}, \boldsymbol{E})$ consisting of a node set $\boldsymbol{V}$ representing random variables with some distribution $\mathbb{P}(\boldsymbol{V})$ and an edge set $\boldsymbol{E}$ consisting of directed edges ($\rightarrow$) and bidirected edges ($\leftrightarrow$) between pairs of different nodes. Node pairs connected by an edge of any type are *adjacent*, and a sequence of alternating nodes and edges, ending with a node, is a *walk*. A *path* is a walk where every node occurs at most once. A *collider* is a node $X$ along a path with edge configuration $\ast\!\!\rightarrow X \leftarrow\!\!\ast$, where an asterisk indicates that an endpoint is either an arrowhead or a tail.

In the canonical causal interpretation, directed edges represent *direct* causal relations with respect to $\boldsymbol{V}$ and bidirected edges represent latent confounding, and can be viewed as originating from an underlying graph where the confounding variables have been marginalized out. We say that $X$ is a *direct cause* and *parent* of $Y$ if there exists a directed edge from $X$ towards $Y$. An *ancestral* relation from $X$ to $Y$ is when there exists a path between $X$ and $Y$ where all edges are directed and with an arrowhead towards $Y$. We then refer to $X$ as an *ancestor* of $Y$, and $Y$ as a *descendant* of $X$. The sets of parents / ancestors / descendants of $X$ are denoted as $\text{PA}(X)$ / $\text{AN}(X)$ / $\text{DE}(X)$ respectively, where the same notation is used for sets $\boldsymbol{X}$ disjunctively.

A *Markov property* relates separation properties of the graph to conditional independences in the corresponding distribution. As we allow for cycles to be present, the $\sigma$-separation Markov property is a non-trivial generalization of the $d$-separation Markov property (Forré and Mooij, 2017), see Appendix A. A causal graph is often associated to a *Structural Causal Model* (SCM), where equations represent causal mechanisms as functions of observed endogenous variables and independent exogenous noise variables, see e.g. Pearl (2009). Effectively, DMGs with the $\sigma$-separation property are a graphical representation of *Simple* SCMs, a convenient subclass of all cyclic SCMs that allows for certain cyclic interactions while maintaining intuitive causal semantics (Bongers et al., 2021).

## 2.2. Interventional Data and Background Knowledge

We use the *Joint Causal Inference* (JCI) framework (Mooij et al., 2020) to model certain background knowledge, interventions and other manipulations with *context variables* $\boldsymbol{C} \subset \boldsymbol{V}$, where the remainder are denoted as *system variables* $\boldsymbol{V} \setminus \boldsymbol{C}$. The central aim of JCI is to define a meta-system of combined system and context variables that can be used for a variety of causal methods.

The full JCI framework requires up to four assumptions, of which we use the following here:

**JCI assumption 0** (Joint SCM) *The data-generating process underlying system and context variables is modeled by a single simple SCM.*

**JCI assumption 1** (Exogenity) *No system variable is an ancestor of any context variable.*

We refer to this combined modeling assumption as JCI-1. For the remainder of this work we have at most one context variable, denoted as $C$, which is graphically represented by a square node. We direct the reader to Mooij et al. (2020) for the JCI assumptions related to larger context models that are outside the scope of this work.

### 2.3. Selection Bias Variables

The presence of selection bias, represented by variable set $\boldsymbol{S}$, can severely hinder statistical procedures and the estimation of causal effects. It affects conditional distributions within the set of observed variables $\boldsymbol{V}$ through a generally unknown mechanism, which in principle could make the estimation of a causal effect strength arbitrarily difficult. In constraint-based causal discovery however, the target is the identification of a causal relation, establishing its presence or absence by reasoning from conditional independence constraints. The presence of $\boldsymbol{S}$ in general alters the independence model of observed set $\boldsymbol{V}$. In this work, the main task is the discovery of ancestral relations, as opposed to estimating their causal effect strength.

We primarily use directed mixed graphs, which are not closed under conditioning, and where we explicitly include each selection bias variable $S \in \boldsymbol{S}$ (as opposed to ancestral graphs, see Richardson and Spirtes (2002)). Here, $\boldsymbol{S}$ encodes for each realization whether that sample is included in the data through an (unobserved) mechanism $\mathbb{P}(\boldsymbol{S} \mid \boldsymbol{V})$. Graphically, a selection bias variable is represented by a shaded and conditioned node (as in Fig. 1), to emphasize its hidden and conditioned state.

## 3. Constraint-Based Causal Discovery

We are interested in constraint-based causal discovery, inferring causal relations from conditional (in)dependence signals in data that is biased through selection. More precisely, the aim is to identify the presence of ancestral causal relations when the underlying causal graph and the selection bias mechanism are unknown.

We assume $\sigma$-*faithfulness* (which we will refer to as 'faithfulness' henceforth), implying that $\sigma$-separations in the graph explain all conditional independences in the observed distribution. Combined with the Markov property, conditional independences in the data map one-to-one to instances of $\sigma$-separation in the causal graph.

The remainder of this section first introduces logical rules for causal inference. We then show how the LCD method (Cooper, 1997) behaves under selection bias, and that there are no three variable constraint-based methods for causal discovery under selection bias. Lastly, the four variable Y-Structure approaches (Mani et al., 2006; Mooij and Cremers, 2015) are shown to be sound in predicting causal relations when selection bias is admitted.

### 3.1. Logical Causal Inference Rules

In Claassen and Heskes (2011), the concept of a *minimal conditional independence* for disjoint sets of variables $\{X\}, \{Y\}, \boldsymbol{W}, \boldsymbol{Z}$ is introduced:[1]

$$X \perp\!\!\!\perp Y \mid \boldsymbol{W} \cup [\boldsymbol{Z}] := (X \perp\!\!\!\perp Y \mid \boldsymbol{W} \cup \boldsymbol{Z}) \wedge (\forall \boldsymbol{Z}' \subsetneq \boldsymbol{Z} : X \not\perp\!\!\!\perp Y \mid \boldsymbol{W} \cup \boldsymbol{Z}'), \qquad (1)$$

where $\boldsymbol{W}$ represents all conditioned variables other than $\boldsymbol{Z}$. Similarly, a *minimal conditional dependence* is defined as:

$$X \not\perp\!\!\!\perp Y \mid \boldsymbol{W} \cup [\boldsymbol{Z}] := (X \not\perp\!\!\!\perp Y \mid \boldsymbol{W} \cup \boldsymbol{Z}) \wedge (\forall \boldsymbol{Z}' \subsetneq \boldsymbol{Z} : X \perp\!\!\!\perp Y \mid \boldsymbol{W} \cup \boldsymbol{Z}'). \qquad (2)$$

---

1. Without faithfulness, the statements in this section hold for graphical notions of minimal $\sigma$-separations and $\sigma$-connections instead of minimal independences and dependences.

Here, the intuition for the minimality of an independence (dependence) given $\boldsymbol{Z}$ is that $\boldsymbol{Z}$ contains exactly those variables that have a meaningful role in separating (connecting) $X$ from $Y$. Indeed, for all proper subsets $\boldsymbol{Z'}$ of $\boldsymbol{Z}$, $X$ and $Y$ are dependent when conditioning on $\boldsymbol{W} \cup \boldsymbol{Z'}$ in (1), and analogously for (2).

The following useful Lemma was introduced by Claassen and Heskes (2011) as part of a reformulation of FCI, mapping minimal (in)dependences to ancestral causal relations, and generalized to DMGs by Mooij et al. (2020). We note that implications similar to (3) were first introduced by Spirtes and Richardson (1996) and Spirtes et al. (1999).[2]

**Lemma 1 (Claassen and Heskes (2011); Mooij et al. (2020))** *Let* $\{X\}, \{Y\}, \boldsymbol{W}, \boldsymbol{Z}$ *be disjoint sets of variables in a DMG of a simple and faithful SCM. Then the following statements hold:*

$$X \perp\!\!\!\perp Y \mid \boldsymbol{W} \cup [\boldsymbol{Z}] \implies \boldsymbol{Z} \in \mathrm{AN}(X \cup Y \cup \boldsymbol{W}), \tag{3}$$

$$X \not\perp\!\!\!\perp Y \mid \boldsymbol{W} \cup [\boldsymbol{Z}] \implies \boldsymbol{Z} \notin \mathrm{AN}(X \cup Y \cup \boldsymbol{W}). \tag{4}$$

Here, (3) states that a minimal independence results in inferring the presence of at least one ancestral relation. However, a minimal dependence for $Z$ in (4) implies that $Z$ is not ancestral to *all* of $X, Y$ and conditioning set $\boldsymbol{W}$ (which can include a selection variable).

### 3.2. Three Variables: LCD

The *Local Causal Discovery* (LCD) (Cooper, 1997) method combines constraints on three variables with specific background knowledge on the causal relations. It allows for latent variables but not selection bias. A *local* approach, a 'pattern search' across all three variable subsets is carried out when more than three variables are included in the data. In a recent formulation, cycles can be present, and the background information is represented in the form of a context variable $C$ subject to JCI-1 assumptions (Mooij et al., 2020).[3]

**Proposition 2 (LCD)** *Let* $\langle C, X, Y \rangle$ *be an ordered tuple of disjoint variables in a DMG of a simple and faithful SCM, where $C$ is a JCI-1 context variable. If*

$$C \perp\!\!\!\perp Y \mid [X] \tag{5}$$

*then $X \in \mathrm{AN}(Y)$, $Y \notin \mathrm{AN}(X)$, $X$ and $Y$ are unconfounded and $\mathbb{P}\left(Y \mid \mathrm{do}(X)\right) = \mathbb{P}\left(Y \mid X\right)$.*

The proof is straightforward and given in Appendix B. All DMGs that adhere to the LCD conditions are summarized in Fig. 2.

When the data is collected under selection bias $\boldsymbol{S}$, the LCD constraints in (5) that can found in the data are essentially replaced with

$$C \perp\!\!\!\perp Y \mid [X] \cup \boldsymbol{S}. \tag{6}$$

As has already been pointed out by Cooper (1997), (6) combined with the LCD background knowledge is not sufficient to identify causal relations. Still, a practitioner might be oblivious to the

---

2. In Spirtes et al. (1999) it can be found as $X \perp\!\!\!\perp Y \mid \boldsymbol{S} \cup [\boldsymbol{W} \cup \boldsymbol{Z}] \implies \boldsymbol{Z} \in \mathrm{AN}(X \cup Y \cup \boldsymbol{S})$, a special case of (3).

3. The original form of LCD is slightly more general with the weaker assumption $X \notin \mathrm{AN}(C)$. The remainder of this section also holds for this formulation.

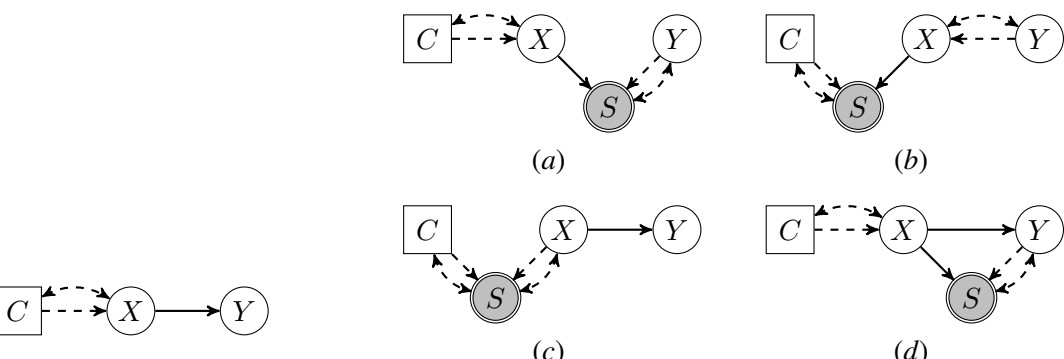

Figure 2: **(LCD, no selection bias)** The three DMGs that adhere to LCD constraints without selection bias. Multiple dashed edges between a pair of nodes indicate that at least one of the edges must be present.

Figure 3: **(LCD, selection bias)** Directed mixed graphs that satisfy the LCD pattern in (6) in addition to Fig. 2 when selection bias $S$ is present. Multiple dashed edges between a node pair indicate that at least one of the depicted edges must be present. Failure modes are presented in (a) and (b), where $X \notin \text{AN}(Y)$, but $X \in \text{AN}(Y)$ in the graphs in (c) and (d).

selection bias mechanism acting on the data at-hand, and we investigate LCD further in this condition.

In Fig. 3, several causal graphs that adhere to (6) are shown. Fig. 3(a) shows a failure mode of (6) where $S$ induces a spurious adjacency between $X$ and $Y$, but where $X$ and $Y$ are non-adjacent without selection bias. Curiously, in Fig. 3(b) we have modes where $Y \in \text{AN}(X)$ in the graph, the reverse of the causal relation to be inferred. On the other hand, Fig. 3(c) depicts an instance where the selection mechanism contributes to the dependence between $C$ and $X$, leading to $X \in \text{AN}(Y)$. This implies that one could find an increase in the number of discovered LCD independence patterns with the correct causal relation from data when a selection bias mechanism is enabled, compared to the same graph where it is absence. In Fig. 3(d) one of the possibilities is shown where $X \in \text{AN}(Y)$, but the size of the causal effect is not estimable.

The result for LCD under selection bias leads to the following impossibility.

**Proposition 3** *When selection bias is present, there exists no three variables constraint-based method for inferring the presence of ancestral causal relations from (in)dependence testing combined with JCI-1 background knowledge.*

We note that it is yet unknown how the partial ancestral graph equivalence class behaves under JCI-1 assumptions, as pointed out by Mooij and Claassen (2020). This prevents a proof strategy that reasons directly over PAGs, and instead we exhaustively search over all DMGs with optional JCI-1 background knowledge. Due to the sheer number of possible DMGs, this enumeration is automated.

### 3.3. Four Variables: Y-Structures

Given the results for three variables, we now turn to the four variable case.

The *Y-Structure* (Mani et al., 2006) and the *Extended Y-Structure* (Mooij and Cremers, 2015) algorithms, which we collectively refer to as 'Y-Structures', do not require the presence of a JCI-1 context variable. Each method searches for a specific pattern of (in)dependence constraints between a quadruple of variables. In Mani et al. (2006) and Mooij and Cremers (2015), it is shown that Y-Structures imply an ancestral causal relation which is unconfounded when the presence of selection bias is excluded. Cooper (1997) stated that the constraints from an Extended Y-Structure imply that selection bias can be excluded, but did not give a proof.

Here we show soundness of the Extended Y-Structure when cycles are allowed and multiple selection bias variables may be present.

**Proposition 4 (Extended Y-Structure)**  *Let $\langle V, W, X, Y \rangle$ be an ordered tuple of disjoint variables in a DMG of a simple and faithful SCM and $\boldsymbol{S}$ be the set of selection bias variables. If*

$$
\begin{aligned}
V &\perp\!\!\!\perp Y \,|\, [X] \cup \boldsymbol{S} \\
V &\not\perp\!\!\!\perp W \,|\, [X] \cup \boldsymbol{S},
\end{aligned}
\tag{7}
$$

*then $X \in \textsc{an}(Y)$, $Y \notin \textsc{an}(X)$, $X \notin \textsc{an}(\boldsymbol{S})$ and $X$ and $Y$ are unconfounded.*

**Proof** First we note that we can assume without loss of generality that there are no other variables in the DMG/SCM than $\{V, W, X, Y\} \cup \boldsymbol{S}$. Indeed, if there were, we could simply marginalize them out.[4] Since ancestral relations are preserved by the marginalization, the inferred ancestral relations in the marginalized graph must also hold in the original graph. Also, marginalization can only add bidirected edges to the remaining variables, and never remove them. Hence the conclusion of unconfoundedness must also hold in the original graph.

The application of Lemma 1 to $V \perp\!\!\!\perp Y \,|\, [X] \cup \boldsymbol{S}$ implies that $X \in \textsc{an}(V \cup Y \cup \boldsymbol{S})$, and applying it to $V \not\perp\!\!\!\perp W \,|\, [X] \cup \boldsymbol{S}$ gives $X \notin \textsc{an}(V \cup W \cup \boldsymbol{S})$, i.e. $X \notin \textsc{an}(\boldsymbol{S})$. Substitution leads directly to $X \in \textsc{an}(Y)$.

We now consider paths between $V$ and $Y$, and note that all such paths must be $\sigma$-blocked given $X \cup \boldsymbol{S}$, as $V \perp\!\!\!\perp Y \,|\, X \cup \boldsymbol{S}$. But there must exist a $\sigma$-open path $\mathcal{P}$ between $V$ and $Y$ given only $\boldsymbol{S}$, as $V \not\perp\!\!\!\perp Y \,|\, \boldsymbol{S}$ in (7). However, $V \perp\!\!\!\perp W \,|\, \boldsymbol{S}$ implies that all paths between $V$ and $W$ are $\sigma$-blocked given $\boldsymbol{S}$, such that $\mathcal{P}$ cannot contain $W$. On the other hand, $\mathcal{P}$ must contain $X$ such that it can be blocked given $X \cup \boldsymbol{S}$. We are left with checking the remaining options for path $\mathcal{P}$.

Suppose $\mathcal{P}$ ends with an edge between $Y$ and some $S_i \in \boldsymbol{S}$ i.e. $V \ldots S_i \ast\!\!-\!\!\ast Y$, where $\ast\!\!-\!\!\ast$ can be any of the edges $\leftarrow$, $\rightarrow$ and $\leftrightarrow$. Then, $\mathcal{P}$ must be of the form $V \cdots \ast\!\!\rightarrow X \leftarrow\!\!\ast S_j \ldots S_i \ast\!\!-\!\!\ast Y$, where the subpath $S_j \ldots S_i$ consists entirely of nodes in $\boldsymbol{S}$, and the orientations near $X$ are due to $X \notin \textsc{an}(V \cup \boldsymbol{S})$. This is a contradiction, because $X$ is a collider that is not ancestor of $\boldsymbol{S}$, hence blocking the path. Thus $\mathcal{P}$ must be of the form $V \ldots X \ast\!\!-\!\!\ast Y$. Since $X \notin \textsc{an}(V \cup W \cup \boldsymbol{S})$, all edges between $X$ and $\{V, W\} \cup \boldsymbol{S}$ are with an arrowhead on $X$, such that $\mathcal{P}$ must be of the form $V \cdots \ast\!\!\rightarrow X \rightarrow Y$, where $X \leftrightarrow Y$ and $X \leftarrow Y$ are excluded as otherwise we would again obtain a contradiction by $X$ being a collider.

Finally, it remains to be shown that $Y \notin \textsc{an}(X)$ and that $X \leftrightarrow Y$ is not in the graph. If either were the case, then it would imply the existence of a $\sigma$-open path between $V$ and $Y$ given $X \cup \boldsymbol{S}$, which would be a contradiction. ∎

---

4. For an acyclic SCM, this can be simply done by substitution, for a cyclic (simple) SCM the operation is somewhat more involved. On the graph level, the marginalization corresponds with an operation also referred to as latent projection. See Bongers et al. (2021) for details on marginalizations.

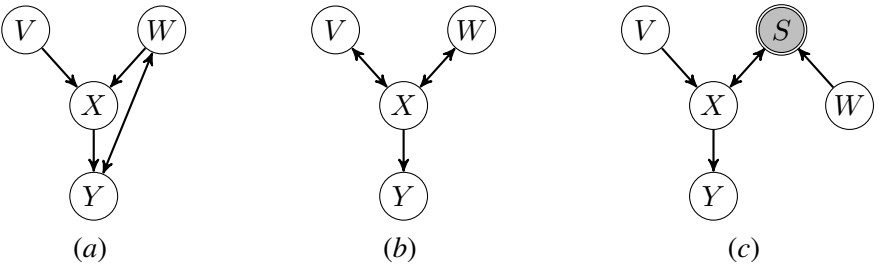

Figure 4: **(Y-Structures)** Examples of mixed graphs that adhere to the Y-Structure (a) and the Extended Y-Structure (b) patterns. In (c), the presence of selection bias $S$ yields an open path between auxiliary variable $W$ and $Y$.

Unconfoundedness of $X$ and $Y$, $X \in \text{AN}(Y)$ and $Y \notin \text{AN}(X)$ together lead to the following.

**Corollary 5** *The causal relation $X \in \text{AN}(Y)$ in Prop. 4 is identifiable, i.e.*

$$\mathbb{P}\left(Y \mid \text{do}(X) \cup \boldsymbol{S}\right) = \mathbb{P}\left(Y \mid X, \boldsymbol{S}\right). \tag{8}$$

The Y-Structure method imposes two more constraints $W \perp\!\!\!\perp Y \mid [X] \cup \boldsymbol{S}$ in addition to (7), symmetrizing the equations for $W$ and $V$ (Mooij and Cremers, 2015). Soundness follows from Prop. 4. Examples of graphs of the (Extended) Y-Structures are depicted in Fig. 4.

## 4. Experiments

We present the results of several simulation experiments, where we apply causal discovery method to biased and unbiased data, and several real-world experiments on gene expressions. Code for the experiments in this section is provided at https://github.com/philipversteeg/sbcd.

### 4.1. Methods and Estimators

Apart from LCD and Y-Structures, we include Invariant Causal Prediction (ICP) (Peters et al., 2016), a state-of-the-art method for causal discovery in this setting (Meinshausen et al., 2016), as a baseline. We denote the practical estimators of LCD, Y-Structures, Extended Y-Structures and ICP as `LCD`, `YSt`, `YSt-Ext` and `ICP` respectively. See Appendix C for details on the ICP method and on the implementation of all practical estimators used.

**Finite Sample Scoring** In practice, each method assigns a score to each discovered causal relation, indicating a level of confidence in the prediction. For `LCD`, we follow Mooij et al. (2020) in using $-\log(p_{CY})$, where $p_{CY}$ is the $p$-value associated under the null hypothesis of independence between context variable $C$ and target variable $Y$.[5] For `ICP` we use the maximum of $p$-values for the predicted parents.

In `YSt` and `YSt-Ext`, for each discovered relation $X \in \text{AN}(Y)$ we compute

$$\max_{V',W'} \min\left(-\log(p_{V'Y}), -\log(p_{W'Y})\right), \tag{9}$$

---

5. Here the $p$-value of any appropriate conditional independence test can be used.

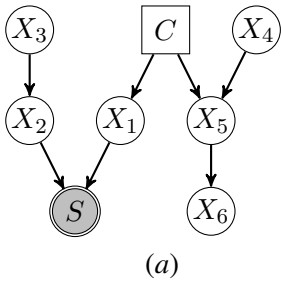

| | **Unbiased Data** $\mathcal{D}_\emptyset$ | | | **Biased Data** $\mathcal{D}_S$ | | |
|---|---|---|---|---|---|---|
| Method | #Pred | TP | FP | #Pred | TP | FP |
| **ICP** | 198 | 197 | 1 | 425 | 200 | 225 |
| **LCD** | 202 | 200 | 2 | 794 | 199 | 595 |
| **YSt–Ext** | 213 | 200 | 13 | 219 | 200 | 19 |
| **YSt** | 198 | 198 | 0 | 200 | 198 | 2 |

(a)             (b)

Figure 5: **(Fixed Graph)** (a) Graph used for fixed-graph simulations, containing a Y-Structure $\langle C, X_4, X_5, X_6 \rangle$ and a failure mode of **LCD** $\langle C, X_1, X_2 \rangle$. (b) Results of applying **ICP**, **LCD**, **YSt** and **YSt–Ext** to the 200 random causal models with the fixed graph in (a). The total number of predicted ancestral relations (#Pred), the number of true positives (TP) and false positives (FP) are given for the unbiased and biased data.

where $p_{V'Y}$ and $p_{W'Y}$ are the $p$-values under the null hypothesis of dependence between $Y$ and both $V'$ and $W'$ respectively, and where we maximize over the discovered patterns $\langle V', W', X, Y \rangle$. A discovered Y-Structure is thus ranked higher when the smallest of the marginal dependences between the target variable $Y$ with both of the two auxiliary variables is larger.

## 4.2. Simulations

We run several experiments where we simulate linear-Gaussian SCMs for both given and randomly sampled directed graphs. In these we include an explicit selection bias node for the preferential sampling and a JCI-1 context node encoding interventional data. As our primary aim in these experiments is assessing algorithm performance under the effects of selection bias, we do not include latent confounding and cycles.

The edge weights between system, context and selection bias variables are sampled uniformly from $[-1.5, -0.5] \cup [0.5, 1.5]$. Weights are rescaled to counter an accumulation of variance among nodes that are further in the topological ordering. The exogenous noise variables are drawn independently from a standard-Gaussian distribution.

**Unbiased and Biased Data** For a given graph and edge weights, two data sets are sampled: one where the selection bias mechanism is present ($\mathcal{D}_S$) and one where it is disabled ($\mathcal{D}_\emptyset$). In $\mathcal{D}_S$, samples are included conditional on $\sum \boldsymbol{S} \in [2, 2.5]$, where the data in $\mathcal{D}_\emptyset$ has no such restriction. This is repeated until 10000 realizations have been accumulated in both data sets.

### 4.2.1. FIXED GRAPH

As a demonstration of the effect of the selection bias mechanism, we sample 200 models with the directed graph in Fig. 5(a). The graph contains a false positive for LCD under selection bias (see Fig. 3(b)) combined with a pattern that adheres to the (Extended) Y-Structure conditions.

The results are shown in Tab. 5(b), where we compare predictions to the true ancestral causal relations in the graph. The true positive count is similar for each method for $\mathcal{D}_\emptyset$, and both **ICP** and **LCD** predict a large amount of false positives for $\mathcal{D}_S$, while **YSt** and **YSt–Ext** show few errors.

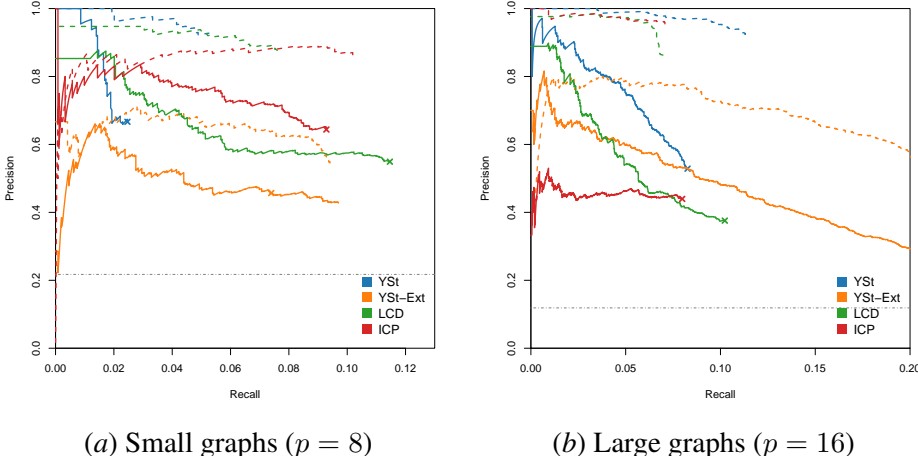

(a) Small graphs ($p = 8$)  (b) Large graphs ($p = 16$)

Figure 6: **(Random Graphs)** PR curves for experiments with random graphs of system size $p = 8$ and $p = 16$. Solid lines indicate performance using data under selection bias ($\mathcal{D}_S$), while dashed lines show the performance for data where the selection mechanism is disabled ($\mathcal{D}_\emptyset$). A cross indicates the threshold where a score that is equivalent to a $p$-value of $0.01$ is reached.

### 4.2.2. RANDOM GRAPHS

We sample small ($p = 8$) and large ($p = 16$) graphs, each including an additional JCI-1 context variable, in a way that promotes spurious correlations due to selection bias (see Appendix D for the procedure).

The results are given in Fig 6. We first note a drop in precision for methods computed on $\mathcal{D}_S$ compared to those using $\mathcal{D}_\emptyset$, indicating a strong effect of introducing selection bias. For the small graphs in Fig. 6(a), `YSt` shows a high precision on a limited recall range, after which `ICP` is outperforming it and other methods. Here `YSt-Ext` is performing considerable worse than `YSt`, which was already pointed out by Mooij and Cremers (2015). In the data without selection bias $\mathcal{D}_\emptyset$, we find that `YSt` is outperforming all others, and is close to `LCD`.

In large graphs (Fig. 6(b)), we see that `YSt` is outperforming all other methods in both $\mathcal{D}_S$ and $\mathcal{D}_\emptyset$ setups. `YSt-Ext` shows a large recall with a drop in precision compare to `YSt`. We find that `ICP` computed with $\mathcal{D}_S$ is considerably less successful when compared to $\mathcal{D}_\emptyset$, a larger difference than found for the small graphs. The large recall of `LCD` might indicate that additional modes as in Fig. 3(c) are created when the selection mechanism is enabled.

We show results for additional experiments in Appendix E, where in one experiment we compare predictions for `LCD`, `YSt` and `YSt-Ext` patterns against the true patterns as existing in the sampled graph, and where we vary the sample size.

### 4.3. Real-World Data

We use real-world microarray data of the yeast genome (Kemmeren et al., 2014). Gene expression levels are captured for each of $p = 6179$ genes under $n = 262$ observational and $m = 1479$ interventional conditions. In each of the latter, one gene has been knocked-out, and expression levels

are measured once for all variables. We combine the samples into a single dataset by adding a binary JCI context variable $C$, where $C = 0$ ($C = 1$) corresponds to all observational (interventional) samples.

We operate here in a high-dimensional setting ($p \gg m > n$), where statistical procedures typically require a form of regularization. Hence, we preselect several variables for `LCD`, `YSt` and `YSt-Ext` using $L_2$-boosting regression, and we reduce the search space of the Y-Structures by fixing the auxiliary variable $V$ in (7) as the context $C$. Details on the practical estimators in this regime are found in Appendix C.3.

### 4.3.1. INTERNAL CROSS-VALIDATION

We use the microarray data in a causal cross-validation setting, where the combined observational and interventional dataset is partitioned into 5 equal parts. In a sequence of 5 experiments, one part is used as a test set and the remaining 4 parts are merged as training data.

The ground truth set of 'true' causal relations is computed from the test set in the following way. Let $X_{j;i}$ be the single sample of the expression of $X_j \in V$ under the intervention of $X_i \in V$, where $i$ and $j$ take values in $[p] = \{1, \ldots, p\}$. Following Versteeg and Mooij (2019), we compute a score $S_{ij}$ representing the size of the absolute effect of a single intervention as found in the data $S_{ij} = \frac{|X_{j;i} - \mu_j|}{\sigma_j}$, where $\mu_j$ and $\sigma_j$ are the empirical mean and empirical standard deviation of the observational test data for $X_j$. The ground truth set for ancestral causal relations between $X_i$ and $X_j$ is then simply $\{(i, j) \in ([p] \times [p]) \mid S_{ij} > t \wedge i \neq j\}$, for some preset value $t$.

In Fig. 7(a), the resulting ROC curve is shown, where $t$ is chosen such that 1 percent of all potential causal relations are contained in the ground truth. We find that `ICP` here is the most successful, while all other methods significantly outperform random guessing (in gray). The `YSt-Ext` method is seen to either outperform or match `LCD` across the range, implying that the using the extra (in)dependence tests involving an additional variable leads to an improvement. We find that `YSt`, which differs from `YSt-Ext` only in the requirement for additional constraints, is seen to perform worse than `YSt-Ext` at all specificity levels. Seemingly, relatively more true positive patterns are discarded by `YSt`, resulting in 96 predicted causal relations compared to a recall of 320 for `YSt-Ext`.

### 4.3.2. DOMAIN VALIDATION

We compile a ground truth of domain knowledge by querying genetic relations from an online compendium of expression data (Cherry et al., 2012) and orienting the causal direction from 'hit' to 'bait', as in Meinshausen et al. (2016). Compared to the internal validation, here we include all data in the training set.

The resulting ROC curves are shown in Fig. 7(b). We find that that `YSt-Ext` outperforms `ICP` and `LCD` over a large portion of the false positive rate. The `YSt` method again performs worse than other methods for most its range. Compared to the internal validation, `YSt` and `ICP` seem less robust than `YSt-Ext`, and `LCD` is performing better in this setting relative to Fig. 7(a).

## 5. Conclusion and Discussion

In this work, we have investigated local constraint-based algorithms when selection bias is present, possibly in addition to latent confounding and cycles. In these conditions, the LCD method was

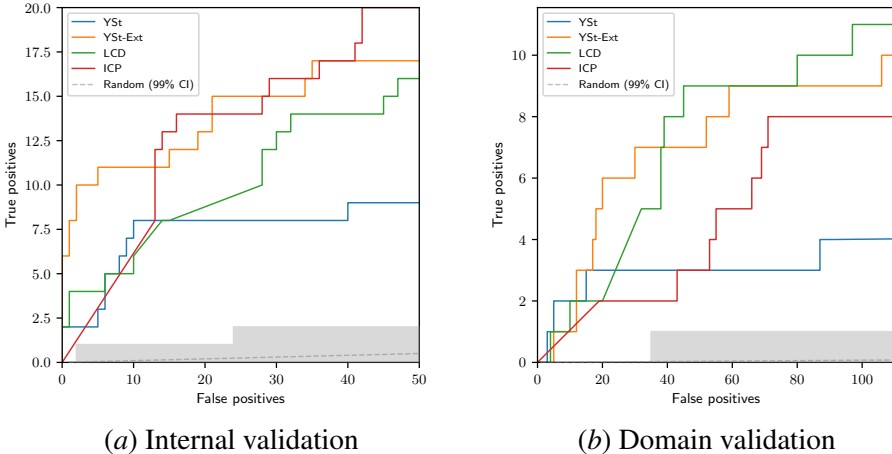

(*a*) Internal validation          (*b*) Domain validation

Figure 7: (**Real-World data**) ROC curves for experiments with real-world microarray data. The gray line represents random guessing and the gray area represents its 99 percent confidence interval.

shown to possibly produce wrong predictions, while also possibly increase recall. Y-Structure type patterns were shown to be sound in predicting ancestral causal relations from data with selection bias. Empirically, we showed that the Y-Structures method works well in some simulation settings using a finite-sample scoring method, while the Extended version has performed poorly. In a real-world setting, we have found that the Extended Y-Structures algorithm is outperformed only by ICP when evaluated against gene expression experiments. There, the Extended Y-Structure variant is shown to be more robust when compared to an external dataset, indicating that the hypothesis of a selection bias mechanism underlying the data is plausible.

Regarding future work, a sound criterion for reading off causal relations from partial ancestral graphs under selection bias is still desired. Meanwhile, our approach, where we investigate small variable sets for soundness under selection bias, can be expanded to higher cardinalities. Approaches such as brute-force searching for valid mixed graphs or a more principled procedure similar to Prop. 4 can also be considered. Finally, the question to what extent background knowledge, such as a JCI assumptions or assumptions on the causal relations between selection variables and other variables, aids in constraint-based causal discovery under selection bias, is left for future work.

## Acknowledgments

PV and JMM are supported by NWO, the Netherlands Organization for Scientific Research (VIDI grant 639.072.410).

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

## Appendix A. $\sigma$-Separation

We first need some additional definitions. A *directed path* from node $X$ to node $Y$ is a path such that all edges on the path are directed and into $Y$. A *directed cycle* is a directed path from $X$ to $Y$ where also $Y \to X$. The *strongly connected component* of $X$ is defined as $\text{SC}(X) = \text{AN}(X) \cup \text{DE}(X)$, and hence contains all nodes on directed cycles that include $X$.

**Definition 6 ($\sigma$-separation (Forré and Mooij, 2017))** *A walk between $X$ and $Y$ in a graph $\mathcal{G} = (\boldsymbol{V}, \boldsymbol{E})$ is $\sigma$-blocked by $\boldsymbol{C} \subseteq \boldsymbol{V}$ if one of the following conditions hold.*

- *$X$ or $Y$ is in $\boldsymbol{C}$.*
- *The walk contains a collider that is not in $\text{AN}(\boldsymbol{C})$.*
- *The walk contains a non-collider $V \in \boldsymbol{C}$ that points to an adjacent node on the walk in another strongly connected component.*

*If all paths between $X$ and $Y$ are $\sigma$-blocked by $\boldsymbol{C}$, then $X$ is $\sigma$-separated from $Y$ by $\boldsymbol{C}$.*

In addition, we refer to a walk or path as *$\sigma$-open* (or just 'open') given $\boldsymbol{C} \subseteq \boldsymbol{V}$ if it is not $\sigma$-blocked by $\boldsymbol{V}$.

## Appendix B. LCD Proof

We give the proof of LCD without select bias.

**Proposition 2 (LCD)** *Let $\langle C, X, Y \rangle$ be an ordered tuple of disjoint variables in a DMG of a simple and faithful SCM, where $C$ is a JCI-1 context variable. If*

$$C \perp\!\!\!\perp Y \mid [X] \tag{10}$$

*then $X \in \text{AN}(Y)$, $Y \notin \text{AN}(X)$, $X$ and $Y$ are unconfounded and $\mathbb{P}(Y \mid \text{do}(X)) = \mathbb{P}(Y \mid X)$.*

**Proof** Equation (5) correspond to a minimal independence (1) for $X$, such that $X \in \text{AN}(C \cup Y)$ by Lemma 1. Now $C$ and $Y$ are $\sigma$-connected because $C \not\perp\!\!\!\perp Y$, but separated given $X$, and thus the path $(C, X, Y)$ forms a non-collider on $X$. Without selection bias, and as $X \notin \text{AN}(C)$, the edge between $C$ and $X$ is either bidirected or directed into $X$, such that the end-point in $X$ is an arrowhead. As we cannot have a collider on $X$ this excludes both $Y \to X$ and $X \leftrightarrow Y$. Thus, the graph must be of the form $C \leftarrow\!\!\!* X \to Y$. This implies that $\mathbb{P}(Y \mid \text{do}(X)) = \mathbb{P}(Y \mid X)$. ∎

## Appendix C. Estimators and Implementation

In general, for pattern-based approaches as **LCD**, **YSt** and **YSt-Ext**, a search for tuples of variables satisfying the relevant constraints is performed over all possible combinations in the variable set. The constraints are checked with a user-specified conditional independence test against a user-specified $p$-value threshold, and, if all hold, the relevant score is computed and returned. The practical estimators for simulations and real-world data are given later in this section, and we first detail the **ICP** baseline method.

### C.1. Invariant Causal Prediction

ICP predicts direct causes $\text{PA}(Y)$ of a target variable $Y$ by testing (in a sophisticated way) if the conditional distribution $\mathbb{P}(Y|\text{PA}(Y))$ remains invariant under changes of an environment (context) variable. In Mooij et al. (2020), ICP is reformulated as predicting ancestral relations by assuming faithfulness. In that formulation, ICP is sound under latent confounding and cycles but not selection bias, similar to LCD.

For the practical estimator **ICP**, we use its standard implementation in the `Invariant-CausalPrediction` package in R with the default parameters. In practice, this results in preselecting a potential parent set with a $L_2$-boosting regression if $p > 8$, and in the application of the following *mean-variance test*, referred to as the 'approximate test' in Peters et al. (2016). It tests for a given potential parent set, for each of the realizations of $c \in C$ of the context (environment) $C$ if the mean of the residuals of a linear regression differs from the mean of the residuals of a linear regression in all other contexts $C \setminus \{c\}$. These $p$-values for all $c \in C$ are combined with a Bonferonni correction. This procedure is repeated for the means of the variances using an $F$-test. Finally, these two $p$-values are also combined with a Bonferonni correction.

### C.2. Estimators for the Simulated Data

In the simulated data we use a standard partial correlation test for **LCD**, **YSt** and **YSt-Ext** where the $p$-value threshold $\alpha$ for rejecting the null hypothesis is set to $\alpha = 0.01$. We 'accept' the null hypothesis of independence for $p$-values above his threshold.

For **ICP**, the standard implementation in the `InvariantCausalPrediction` package requires a discrete context variable, and we discretize $C$ into binary outcomes around its mean value. For the simulations where $p = 16$, we override the default settings so that no preselection is performed.

### C.3. Estimators for the Real-World Data

We use the mean-variance test as described above for testing any (conditional) independence in **LCD**, **YSt** and **YSt-Ext** between the single context variable $C$ and any system variable $\boldsymbol{V} \setminus C$. In the other cases, a standard partial correlation test is used. We use two thresholds, accepting the independence hypothesis for a $p$-value threshold $\alpha$ of 0.01, rejecting the null for a lower threshold of $\alpha/p$, where $p$ is again the number of variables.

**Preselection and High-Dimensionality**  We operate here in a high-dimensional setting ($p \gg m > n$), where statistical procedures typically require a form of regularization.

We use $L_2$-boosting regression (Bühlmann and Yu, 2003) for each target variable $Y$ as a preselection to **LCD** (Versteeg and Mooij, 2019). Here, up to 8 variables are selected by applying the `GLMBoost` routine in the `MBoost` package in R. Essentially this reduces the search for each $X \in \boldsymbol{V}$ in the LCD triple $\langle C, X, Y \rangle$ to a potential parent set $X \in \boldsymbol{V}_Y^{\text{sel}} \subsetneq \boldsymbol{V}$, where $\boldsymbol{V}_Y^{\text{sel}}$ is the set of covariates selected by a $L_2$-boosting regression for $Y$. The **ICP** implementation uses a similar preselection technique. For the Y-Structures estimators, we fix the auxiliary variable $V$ to be the single context variable $C$. Similar to **LCD**, we reduce the search space of both $X$ and $W$ in each $\langle C, W, X, Y \rangle$ pattern to the preselected variable sets $X \in \boldsymbol{V}_Y^{\text{sel}}$ and $W \in \boldsymbol{V}_X^{\text{sel}}$ respectively.

To further improve stability of the predictions in this setting (Meinshausen and Bühlmann, 2010), we bootstrap each method for 100 random subsamples of the data, we use average of the score over all bootstrap samples as the final estimator.

## Appendix D.  Sampling Random Graphs

We sample random graphs with small graphs with $p = 8$ and large graphs with $p = 16$ system variables, and one additional context variable each. The following procedure is repeated for different random seeds until a graph is found.

The directed edges between each node pair are sampled independently with a fixed probability of $0.15$ for $p = 8$ and $0.09$ for $p = 16$. Cyclic graphs and graphs that do not meet a predetermined minimum number of collider patterns dependent are discarded, to increase the prevalence of spurious correlations due to selection bias. We set this parameter to 3 for $p = 8$ and 5 for $p = 16$. For the parents of the selection bias variable, we uniformly sample 1 and 3 variables for $p = 8$ and $p = 16$ respectively from the set of leafs of descendants of all colliders nodes. Finally, we randomly pick with uniform weight one of the source nodes of the graph as a context variable.

## Appendix E.  Additional Random Graph Experiments

Here we include two more experiments on small ($p = 8$) and large ($p = 16$) random graphs, extending the results in Sec. 4.2.2.

### E.1.  Oracle Patterns

We first show results for an experiment where we compare to a ground truth of oracle independence patterns. For each predicted causal relation produced by one of the **LCD**, **YSt** and **YSt−Ext** estimators, we check for each of its associated patterns (which may be multiple patterns for each predicted causal relation) if that independence pattern exists in the true graph. The existence of such a pattern is used as the positive condition in the construction of the PR curve, shown in Fig. 8. Here the score (9) is used to score predictions, without taking the maximum over all discovered patterns. For small graphs, we find that for $\mathcal{D}_S$, highly confident predictions as produced by **YSt** are often true patterns found in the true graph, indicated by a high precision at the top of the ranking for a low recall. For $\mathcal{D}_\emptyset$ it is similar in precision levels to **LCD**. For graphs with $p = 16$, for only the lowest recall **YSt** outperforms **LCD**, after which **LCD** does better. In both sets of graphs, **YSt−Ext** shows by far the worst performance.

### E.2.  Varied Sample Size

We perform experiments with random graphs, which are described in Sec. D, where we vary the total sample size $n$. In Fig 9, PR curves for experiments with small random graphs ($p = 8$) and large random graphs ($p = 16$) are found. In each row, the number of samples $n$ are varied, ranging between $n = 1000$ and $n = 20000$.

Generally, **LCD** performs relatively better at smaller $n$, and the precision of **YSt** and **YSt−Ext** increases with sample size. We find that overall **YSt** shows the worst results, but it performs better on $\mathcal{D}_S$ than on $\mathcal{D}_\emptyset$ for $n = 1000$ at $p = 6$, while this effect reverses for the other cases. **YSt−Ext** outperforms **LCD** in most cases at small recall, and this effect is stronger for larger samples.

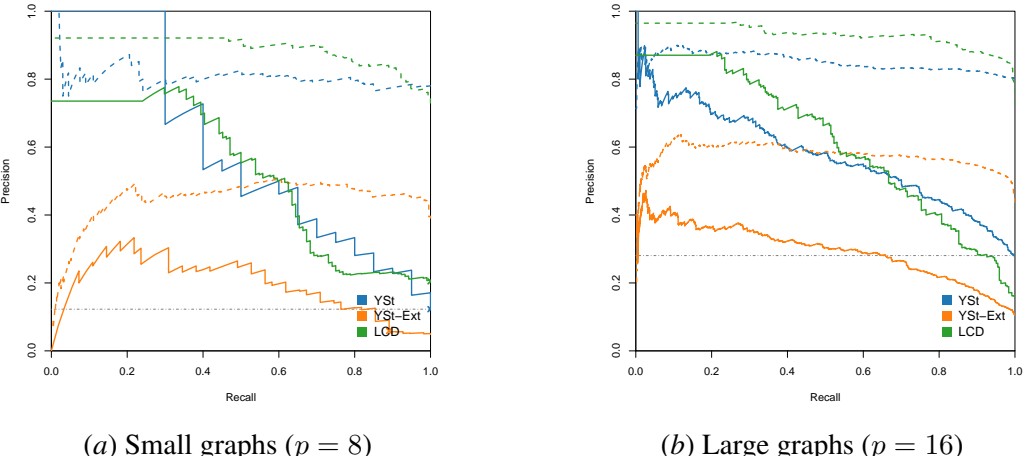

(a) Small graphs ($p = 8$)  (b) Large graphs ($p = 16$)

Figure 8: **(Random Graphs, Oracle Patterns)** PR curves for experiments with random graphs of system size $p = 8$ and $p = 16$, where the condition positive is if the oracle conditional independence pattern found in the true graph. Solid lines indicate performance using data under selection bias ($\mathcal{D}_S$), while dashed lines show the performance for data where the selection mechanism is disabled ($\mathcal{D}_\emptyset$).

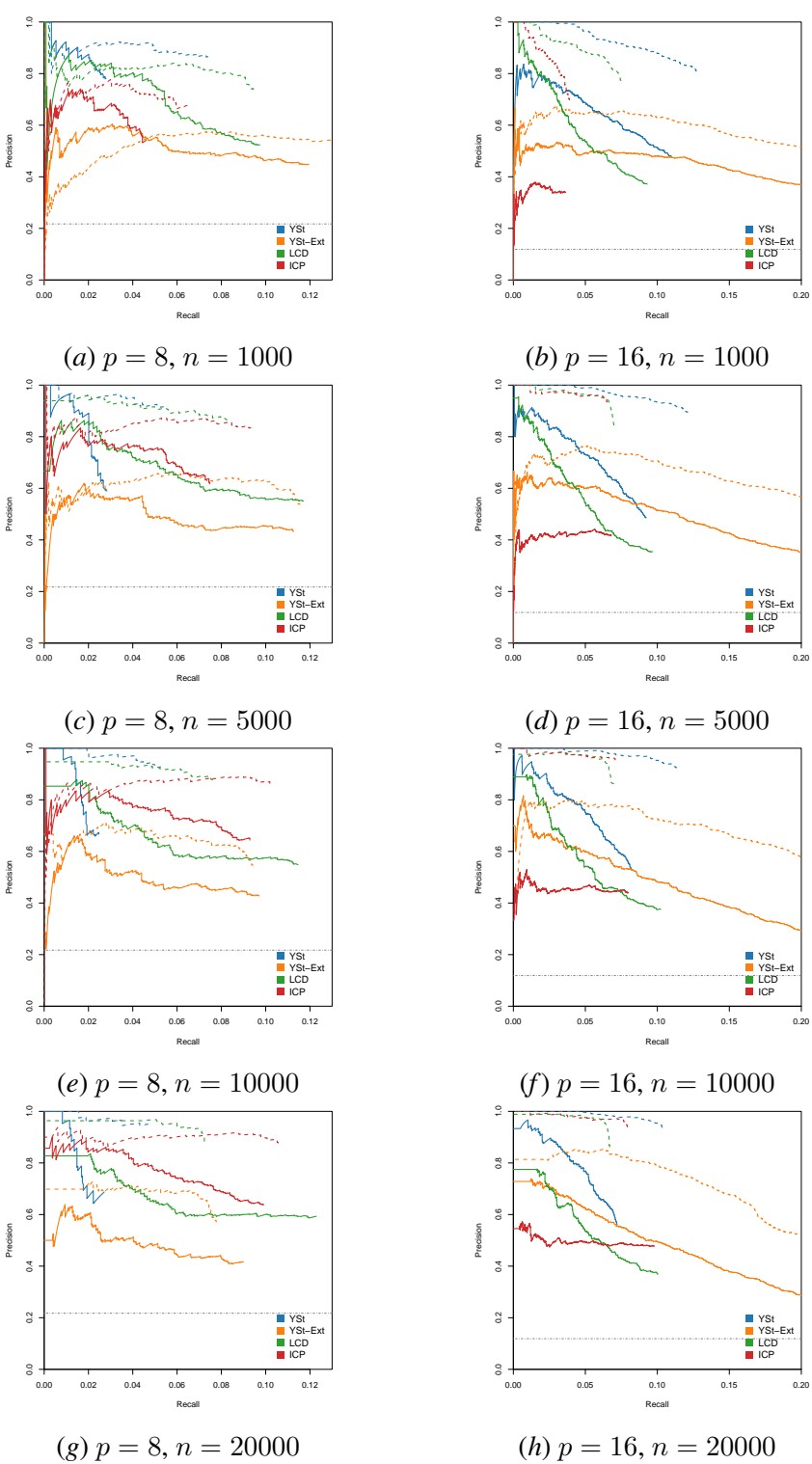

(a) $p = 8, n = 1000$

(b) $p = 16, n = 1000$

(c) $p = 8, n = 5000$

(d) $p = 16, n = 5000$

(e) $p = 8, n = 10000$

(f) $p = 16, n = 10000$

(g) $p = 8, n = 20000$

(h) $p = 16, n = 20000$

Figure 9: **(Random Graphs, varied** $n$**)** PR curves for experiments with small ($p = 8$) and large ($p = 16$) random graphs for various sample sizes. Solid lines indicate performance using data under selection bias $\mathcal{D}_S$, while dashed lines show the performance for data where the selection mechanism is disabled $\mathcal{D}_\emptyset$.

