# OpenReview forum: "Local Constraint-Based Causal Discovery under Selection Bias"
_cclear.cc/CLeaR/2022/Conference — CLeaR 2022 Poster_

### Official Review · Reviewer_5ZRb · 2021-11-20

**Confidence:** 4
**Overall Score:** 6

**Main Review:**


The paper investigates identifying causal (ancestral) relationships among two variables locally. The impossibility result with three-variable case seems new but the result is rather straightforward (due to a small number of possible structures.) For the four-variable case (extended Y-structure), the result can be derived by Lemma 1 (Claassen and Heskes 2011). (One can refine further to X \in An(Y)\An(W).) Given that the mathematical rigor to prove the result is not high, the quality of the paper would be mostly determined by the significance of the question, relevance, and empirical results.

The question seems important that under which condition can we tell ancestral relationship under selection bias. But the question is limited to three and four variable cases. How hard would it be to solve 5-variable case? (certainly one can consider a brute-force approach)

The paper is clearly written but can be improved by
- skipping the discussing of MAG (For example, Figures 2 and 3 are possible ADMGs not MAGs. We do not need MAG to understand the results)
- introducing JCI framework a little bit (in assumption 0 you didn’t define C and X yet)
- draw Y-structure in Section 3.3

Empirical validation is interesting but also can be improved. The current result using random graphs can be quite confusing since non-sound method associates with higher precision and recall even under the selection bias.
- Tests on random graphs only check 1) how many portions of pairs are adequate for ext Yst & 2) how good the estimator is. Hence, it is not quite appropriate to juxtapose PR curves for different methods. They will find orientation for different edges where some might be irrelevant to selection bias.
- Can you report what PR value would be fore alpha=0.05 instead of -log(p) value?

What’s max in Eq. (9)? max_X? also what’s An(sY) just before the equation?

Overall, I like the simple question and solution no one asked before. The presentation and experiments can be improved. To highlight the importance of ext Y-st, testing on more data sets would be desirable although the soundness of the method is already proven.

minor.
AMDG just before section 2.2
In section 4.2.2, “Fig 4.2.2” should be Fig 5
Figure 5, please explicitly mention what are solid (S=0) and dashed (S=1) lines.
In conclusion, “LCD was found to not be sound“ may be written in a different way, e.g., “LCD is not applicable to a data with selection bias”. LCD is sound under the condition it was proposed.


**Summary:**

This paper considers local causal discovery of three or four variables under possible selection bias. With selection bias, the paper presents the impossibility results for determining the ancestral relationship among three variables. When four variables are given, it may be possible to disambiguate ancestral relationship between two variables through simple conditional independence tests. Empirical validation result demonstrates the usefulness in a real-world data set.

---

> ### Author Response · Authors · 2021-12-03
> **We thank the reviewer for the insightful and helpful comments.**
>
> We thank the reviewer for the insightful and helpful comments.
>
> > For the four-variable case (extended Y-structure), the result can be derived by Lemma 1 (Claassen and Heskes 2011). (One can refine further to $X \in An(Y) An(W)$.) Given that the mathematical rigor to prove the result is not high, the quality of the paper would be mostly determined by the significance of the question, relevance, and empirical results.
>
> We have since extended the proposition and proof for the camera ready to 1) allow for cycles in the data and 2) to exclude confounding of the predicted ancestral relation, novel results when selection bias is present. Lemma 1 is not sufficient for proving this, and we need to consider mixed graphs explicitly here.
>
> > But the question is limited to three and four variable cases. How hard would it be to solve 5-variable case? (certainly one can consider a brute-force approach)
>
> Given that the number of mixed graphs is super-exponential in variable count, we expect that the number of distinct sound patterns will grow significantly for the five variable case compared to four variables. We expect a large fraction of these to be straightforward extensions of (Extended) Y-Structures, where the additional freedom gained by an extra variable is not utilised. A brute-force enumeration would be a tractable way of approaching such a search. Additionally, the question of exploiting background knowledge (such as a JCI assumption or otherwise) remains open for four variables. We have now incorporated all these points in the discussion as future work.
>
> > - skipping the discussing of MAG (For example, Figures 2 and 3 are possible ADMGs not MAGs. We do not need MAG to understand the results)
> > - introducing JCI framework a little bit (in assumption 0 you didn’t define C and X yet)
> > - draw Y-structure in Section 3.3
>
> We agree with the reviewer that the inclusion of MAGs is redundant, and we removed this from the manuscript. We have now also simplified and generalised the main aim of this work to the discovery of ancestral relations of an underlying mixed graph, allowing for the presence of cycles. We have improved the JCI section, including definitions of $C$ and $X$ prior to the assumptions. We added a graph of a Y-Structure and an Extended Y-Structure, however, as we now removed MAGs (and PAGs), these are instances of mixed graphs.
>
> > Hence, it is not quite appropriate to juxtapose PR curves for different methods. They will find orientation for different edges where some might be irrelevant to selection bias.
>
> We are not sure why a juxtaposition of PR curves of different methods is inappropriate here. If the reviewer can clarify we can incorporate an improved analysis in a final version.
> Currently, we included a high-level approach in the simulation experiments rather than a fine-grained analysis of e.g. which causal predictions are due to a spurious correlation in the data. The aim here was to adhere to an evaluation one would use in real-word experiments, such as PR and ROC measures, where we switch out a data-driven or domain ground truth to the set of true ancestral relations from the simulator graph.
>
> > Can you report what PR value would be fore alpha=0.05 instead of -log(p) value?
>
> We will report this PR value for each method in the final version.
>
> > What’s max in Eq. (9)? $max_X$? also what’s An(sY) just before the equation?
>
> The maximum is over all discovered (Extended) Y-Structures, we thank the reviewer for pointing out this omission and added it to the manuscript. The $An(sY)$ is a typo for $An(Y)$, which we fixed in the paper.
>
> > minor. AMDG just before section 2.2 In section 4.2.2, “Fig 4.2.2” should be Fig 5 Figure 5, please explicitly mention what are solid (S=0) and dashed (S=1) lines. In conclusion, “LCD was found to not be sound“ may be written in a different way, e.g., “LCD is not applicable to a data with selection bias”. LCD is sound under the condition it was proposed.
>
> We fixed the erroneous figure reference, clarified that the fixed and dashed lines represents biased/unbiased samples and rewrote the referred sentences.

---

> > ### Comment · Reviewer_5ZRb · 2021-12-12
> > **.**
> >
> > Thanks for the response.
> >
> > Regarding the juxtaposition of PR curves, my concern is partly related to hi8A's question and your response ("This is a good point, it could e.g. be a feature of the simulation setup we used, but we are not entirely sure why. We will investigate and add our findings to the experimental section or the discussion.").
> >
> > There are different edges different methods can orient. Even though we are looking at the level of precision for a fixed recall, we are not comparing the orientation of the same edges. Hence, if you, for example, can limit precision/recall only for the edges those that can theoretically be orientable by Yst(-ext), then we can get more refined PR-curves tailored to Yst(-ext).

---

> > > ### Author Response · Authors · 2021-12-23
> > > **Thanks for this clarification.**
> > >
> > > Thanks for clarifying with the specific example. We will add a more fine-grained analysis, taking into account the oracle performance of a method, in addition to the current evaluation to a final version of the paper.

---

### Official Review · Reviewer_KALh · 2021-11-21

**Confidence:** 3
**Overall Score:** 7

**Main Review:**

Locally discovering causal relations is important in practice. Assuming the existence of selection biases and hidden confoundings, this paper investigates whether the LCD algorithm and the extended Y-structure are still valid.

This paper is well-written and the contribution is clear. The authors proved that:

1. For the three-variable case, there is no sound constraint-based approach to discover a purely ancestral causal relation under selection bias.

2. For the four-variable case, the (extended) Y-structure is still sound under selection bias.

However, it seems that the above two results are direct consequences of the logical causal inference rules. The logical causal inference rules (or other similar results) have been used to prove the correctness of RFCI (see, e.g. Colombo et al., 2012, Lemma 3.1). It
would be better to compare the proved results to Lemma 3.1 and to how Lemma 3.1 is applied in RFCI.

Other comments.
1. Section 4.1. It seems that $X\in AN(sY)$ should be $X\in AN(Y)$, and that $X\in Y$ should be $X\in AN(Y)$.
2. In Section 4.1, the authors define a score in Eq.(9). Does this score only work for the linear-Gaussian case? If the variables are discrete, can we still use this score?


**Summary:**

This paper proves that the LCD algorithm fails to discover the causal relation under selection bias, while the extended Y-structure is shown to be sound.

---

> ### Author Response · Authors · 2021-12-03
> **We thank the reviewer for the thoughtful comments.**
>
> We thank the reviewer for the thoughtful comments.
>
> > However, it seems that the above two results are direct consequences of the logical causal inference rules.
>
> We have since expanded the proposition on Y-Structures to include unconfoundedness of the predicted causal relation, while also allowing for cycles to be present in the data. The proof for this does not follow directly from applying logical inference rules, and requires reasoning over mixed graphs.
>
> > The logical causal inference rules (or other similar results) have been used to prove the correctness of RFCI (see, e.g. Colombo et al., 2012, Lemma 3.1). It would be better to compare the proved results to Lemma 3.1 and to how Lemma 3.1 is applied in RFCI.
>
> We have now updated the main text to include references to Spirtes and Richardson, 1996 ("A polynomial time algorithm for determining DAG equivalence in the presence of latent variables and selection bias") and Spirtes et al., 1999 ("An algorithm for causal inference in the presence of latent variables and selection bias"), adding a comment that this is (to the best of our knowledge) the earliest prior work with a formulation similar to the logical causal inference rules, where it is used in proving soundness of FCI. We do not see how comparing the logical causal inference rules to the application of Lemma 3.1 in RFCI is related to the setting of this work.
>
> > Section 4.1 It seems that $X \in AN(sY)$ should be $X \in AN(Y)$, and that $X \in Y$ should be $X \in AN(Y)$
>
> We thank the reviewer for pointing out these typos, we fixed this in the paper.
>
> > In Section 4.1, the authors define a score in Eq.(9). Does this score only work for the linear-Gaussian case? If the variables are discrete, can we still use this score?
>
> One can use the p-value of any appropriate conditional independence test, e.g., the G test for discrete variables, for the score in Eq.(9). We added a comment to this extend to the manuscript.

---

> > ### Comment · Reviewer_KALh · 2021-12-19
> > **Thanks for the reply**
> >
> > I thank the authors for the responses, and I am happy to raise my score to 7.

---

### Official Review · Reviewer_hi8A · 2021-11-22

**Confidence:** 4
**Overall Score:** 7

**Main Review:**

Originality:

The main contributions of the paper are its presentation of simple, elegant proofs of consistency for three-variable (LCD) and four-variable (Y-structure) causal discovery algorithms. The proof for the three-variable algorithm shows consistency under hidden confounding. The proof for the four-variable algorithms shows consistency under both hidden confounding and selection bias.

The current paper discusses earlier work on the three-variable LCD algorithm and points to a paper from 1997 that describes it. A scan of that paper indicates that it explicitly assumes the absence of selection bias (Assumption 5, page 207). It seems important that the current paper make clear that the earlier paper was explicit in describing this assumption; otherwise, the reader of the current paper may be left with the impression that the earlier work was unaware of this assumption, and thus, so may be the readers of that earlier paper. Also, the earlier LCD paper describes how Y structures can be used to exclude the possibility of selection bias (page 217), although it does not provide a consistency proof. It seems important that the current paper mention that Y structures were introduced in the earlier paper as a way to discover causal relationships that have no hidden confounding or selection bias.

Significance:

The computational tractability of local causal discovery algorithms makes them appealing. The relatively few conditional independence tests required to perform causal discovery with three-variable and four-variable models make them potentially more statistically reliable than global causal discovery algorithms, such as FCI, in which the correctness of the causal relationships in the single graph that is output can be highly dependent on one another. However, what about the need to control for multiple testing when evaluating thousands or even millions of three- and four-variable models when performing local causal discovery?

A natural question is this: What might be gained by performing local causal discovery using more than four variables? It seems worthwhile for the paper to at least touch on this issue about future research in the Discussion section.

Technical quality:

The technical quality of the paper is excellent. The proofs appear to be correct.

The definitions of minimal conditional independence and dependence in Section 3.1 appear to be different than the definitions given in the cited paper by Claasen and Heskes (2011). The definitions given by Claasen and Heskes seem to make more sense. The authors should check to make sure that their definitions are the same as those given in the Claasen and Heskes paper.

The experiments include both simulated and real data, and they provide useful insights about the algorithms. It would be helpful to have more details about how the random graphs were generated. For example, directed edges were sampled independently with some fixed probability, which is not stated in the paper. Also, graphs that did not meet a predetermined minimum number of collider patterns were discarded, but that number is not stated in the paper. The paper generated 10,000 samples for the random graph experiments. It would be interesting to see how the results change over a wide span of sample sizes. For the large random graphs, it is interesting that the maximum precision (over all levels of recall) of YSt-Ext is lower than LCD and YSt. Why is that? It would be useful if the paper analyzed statistically the differences in results, using hypothesis testing or confidence intervals, for example. The discussion and interpretation of the experimental results is sparse; additional discussion would be useful.

Clarity:

The paper is well written and easy to read, overall. There are a few places that would benefit from clarification, and there are a few minor typos:

* Example 1 is a bit confusing. For instance, how does the professor observe spurious improvement in the final grades G of those who handed in their exams, if “only students with good grades actually handed-in the class review sheet, which is privacy-sensitive and unobservable to the professor”? Is it the case that the professor does not know the content of the review sheets, but does know which students hand-in the review sheets? It would be interesting if the example was worked out numerically (outside of the paper) and the direction of the final results reported in the paper.

* Figure 1. It would be good to mention in the caption what each of the node types represents.

* Example 2. In the phrase “might destroy some of these genes” it seems the word “genes” should be “cells.”

* “In fact, Fig. 2 represents” should be “In fact, Fig. 1b represents”

* It would be helpful to have a more expanded explanation of the following sentence on page 2: “… a procedure for extracting (non)ancestral relations from this equivalence class when selection bias is a possibility is yet unknown.”

* Page 3. “valid under when”  “valid when”

* Page 3. “Constraint-based method causal discovery methods”  “Constraint-based causal discovery methods”

* Page 3. “between nodes relates to an”  “between nodes to an”

* Page 4. It seems the phrase “a conditioned variable that is marginalized out” should be “a variable that is conditioned on.”

* Page 4. “Anterior causal relations” is mentioned, but not defined or described.

* Page 4. Assumption 0. C and X need to be defined or described.

* Page 4. Mooij et al. (2020) is an ambiguous citation, because it matches two of the references.

* Page 5. The phrase “where W represents selection all other conditioned variables” is not clear.

* Page 6. Shouldn’t the phrase “Z is not ancestral to all of X, Y and conditioning set W” be “Z is not ancestral to any of X, Y and conditioning set W”?

* Page 6. “Soundness follows of the first two”  “Soundness of the first two.”

* Page 6. Shouldn’t “the original assumption of LCD is slightly stronger, only requiring” be “the original assumption of LCD is slightly weaker, only requiring”?

* Page 8. It would be clarifying if Section 3.3 contained a diagram of a Y structure.

* Page 8. “Extended Y-Structure.(7)”  “Extended Y-Structure (7)”

* Page 9. The section of “Finite Sample Scoring” is not easy to follow.

* Page 10. What is “Fig. 4.2.2”?

* Page 11. Figure 5. It would be helpful if the caption describes the meaning of solid and dotted lines in the plots.

* Page 12. The meaning of and reasoning underlying the following sentence is not entirely clear to me: “In Fig. 6(a), the resulting ROC curve is shown, where the threshold t is chosen in [a] way that 1 percent of the causal relations is contained in the ground truth.” Why choose 1 percent, and what are the likely repercussions for doing so?


**Summary:**

This paper focuses on constraint-based causal structure discovery of ancestral relationships from observational data when using only three or four variables at a time, which is termed local causal discovery. The paper describes conditions in which three variables are sufficient to discover (from observational data alone) ancestral causal relationships that have no hidden confounding. It also describes that three variables cannot exclude the possibility of selection bias, whereas local discovery using four variables sometimes can. Evaluations using synthetic and real data provide some support for results that are expected from the theory, although the results are more nuanced than perhaps anticipated.

---

> ### Author Response · Authors · 2021-12-03
> **We thank the reviewer for the detailed comments.**
>
> We thank the reviewer for the detailed comments.
>
> > The current paper discusses earlier work on the three-variable LCD algorithm and points to a paper from 1997 that describes it. A scan of that paper indicates that it explicitly assumes the absence of selection bias (Assumption 5, page 207). It seems important that the current paper make clear that the earlier paper was explicit in describing this assumption; otherwise, the reader of the current paper may be left with the impression that the earlier work was unaware of this assumption, and thus, so may be the readers of that earlier paper.
>
> Good point, we added this clarification to the manuscript.
>
> > Also, the earlier LCD paper describes how Y structures can be used to exclude the possibility of selection bias (page 217), although it does not provide a consistency proof. It seems important that the current paper mention that Y structures were introduced in the earlier paper as a way to discover causal relationships that have no hidden confounding or selection bias.
>
> We thank the reviewer for pointing out this passage from prior work, this is relevant to this paper that we have now included.
>
> > However, what about the need to control for multiple testing when evaluating thousands or even millions of three- and four-variable models when performing local causal discovery?
>
> This is why we rank the patterns found according to a score that expresses the strength of the conditional dependence(s) in terms of the $-\log p$ values. Coming up with more principled solutions is challenging because we (as constraint-based methods generally do) depend both on low type I error and low type II error, and is considered future work (the only causal discovery algorithm we know of that addresses this somewhat satisfactorily is the ICP algorithm).
>
> > A natural question is this: What might be gained by performing local causal discovery using more than four variables? It seems worthwhile for the paper to at least touch on this issue about future research in the Discussion section.
>
> We have now added a comment on this to the future work in the discussion section, where we will consider 1) adding a fifth (or more) model variable and 2) incorporating background knowledge such as JCI to the four variable case.
>
> > The definitions given by Claasen and Heskes seem to make more sense. The authors should check to make sure that their definitions are the same as those given in the Claasen and Heskes paper.
>
> This is an error in our definitions, we have fixed this in the manuscript.
>
> > It would be helpful to have more details about how the random graphs were generated. For example, directed edges were sampled independently with some fixed probability, which is not stated in the paper. Also, graphs that did not meet a predetermined minimum number of collider patterns were discarded, but that number is not stated in the paper.
>
> We have now updated the manuscript to include these left-out parameters of the simulator.
>
> > The paper generated $10,000$ samples for the random graph experiments. It would be interesting to see how the results change over a wide span of sample sizes.
>
> We will add results for simulation experiments over a range of sample sizes to the appendix.
>
> > For the large random graphs, it is interesting that the maximum precision (over all levels of recall) of YSt-Ext is lower than LCD and YSt. Why is that? It would be useful if the paper analyzed statistically the differences in results, using hypothesis testing or confidence intervals, for example. The discussion and interpretation of the experimental results is sparse; additional discussion would be useful.
>
> This is a good point, it could e.g. be a feature of the simulation setup we used, but we are not entirely sure why. We will investigate and add our findings to the experimental section or the discussion.
>
> > The paper is well written and easy to read, overall. There are a few places that would benefit from clarification, and there are a few minor typos:
>
> We thank the reviewer for thoroughly reading and pointing out these errors, which we corrected in our manuscript. We have now reworked the examples in the introduction to be 1) more related to experimental setting in the work and 2) to be more clear in pointing out difficulties due to selection bias. We fixed the typos, clarified both figures, added a definition for ancestral causal relations, cleared up the erroneous figure reference and have disambiguated the citation for Mooij et al. (2020). The ground truth prevalence was set to 1 percent as that level has been used in prior work on this data, but the level is arbitrary. We have added a comment on this to the manuscript.

---

> > ### Comment · Reviewer_hi8A · 2021-12-11
> > **Thanks to the authors for their response.**
> >
> > I thank the authors for their helpful response to my comments and questions.

---

### Decision · Program_Chairs · 2022-01-12

**Decision:**

Accept (Poster)

**Comment:**

This paper shows that when making several assumptions of the Joint Causal Inference framework (introducing exogenous context variables), the local causal discovery algorithm is able to reliably discover a causal effect when there are latent variables, but no conditional independence based algorithm can discover a causal effect when there are latent variables and selection bias. It also shows that when using only 4 variables, there is a consistent way to discover a causal effect when there are latent variables, selection bias, and cycles (assuming a simple structural causal model). The failure modes of local causal discovery with three nodes under selection bias, and the ability of extended Y-structures (the 4 variable case) to discover causal effects under latent variables, selection bias, and cycles is new. The paper is generally well written and easy to follow. It is technically correct. The results are of interest because although the results are about rather special cases, there is some evidence that very simple, local algorithms are more reliable than more informative global algorithms.

However, the authors were asked to submit a modified version of their paper during the discussion period. They submitted a new version, with some new results. However, in examining it more closely while writing the meta-review, I found that they had introduced some very minor errors, which need to be fixed. These include:

1. There is a definition of DMGs but not ADMGs.
2. There is no reference to MAGs prior to Assumption 0. Either Assumption 0 should be changed to refer to DMGs, or ADMGs or MAGs should be introduced. Whichever one of these models is assumed, the authors should make certain that it is compatible with the revised 4 variable theorem about cyclic models.
3. The proof of proposition 4 was changed to allow for the theorem to apply cyclic simple SCMs. This is not clearly stated in the theorem, which should make clear that the assumption of a possibly cyclic but simple SCM is allowed. This should also be made clearer in the abstract and introduction.

Two reviewers scored this paper as a 7, and one as a 6.